# Identification of hub genes associated with COVID-19 and idiopathic pulmonary fibrosis by integrated bioinformatics analysis

Qianyi Chen[1], Shilin Xia[2,3]*, Hua Sui[1], Xueying Shi[1,2], Bingqian Huang[1,2], Tingxin Wang[1]

1 Institute (College) of Integrative Medicine, Dalian Medical University, Dalian, Liaoning, China, 2 Clinical Laboratory of Integrative Medicine, The First Affiliated Hospital of Dalian Medical University, Dalian, Liaoning, China, 3 Department of Palliative Medicine, Graduate School of Medicine, Juntendo University, Tokyo, Japan

* xiashilin@dmu.edu.cn

**Data Availability Statement:** All data generated or analyzed during this study are included in this published article and its supplementary information files.

## Abstract

### Introduction

The coronavirus disease 2019 (COVID-19), emerged in late 2019, was caused by severe acute respiratory syndrome coronavirus 2 (SARS-CoV-2). The risk factors for idiopathic pulmonary fibrosis (IPF) and COVID-19 are reported to be common. This study aimed to determine the potential role of differentially expressed genes (DEGs) common in IPF and COVID-19.

### Materials and methods

Based on GEO database, we obtained DEGs from one SARS-CoV-2 dataset and five IPF datasets. A series of enrichment analysis were performed to identify the function of upregulated and downregulated DEGs, respectively. Two plugins in Cytoscape, Cytohubba and MCODE, were utilized to identify hub genes after a protein-protein interaction (PPI) network. Finally, candidate drugs were predicted to target the upregulated DEGs.

### Results

A total of 188 DEGs were found between COVID-19 and IPF, out of which 117 were upregulated and 71 were downregulated. The upregulated DEGs were involved in cytokine function, while downregulated DEGs were associated with extracellular matrix disassembly. Twenty-two hub genes were upregulated in COVID-19 and IPF, for which 155 candidate drugs were predicted (adj.P.value < 0.01).

### Conclusion

Identifying the hub genes aberrantly regulated in both COVID-19 and IPF may enable development of molecules, encoded by those genes, as therapeutic targets for preventing IPF progression and SARS-CoV-2 infections.

**Funding:** The author(s) received no specific funding for this work.

**Competing interests:** The authors have declared that no competing interests exist.

## Introduction

Severe acute respiratory syndrome coronavirus 2 (SARS-CoV-2), a novel enveloped RNA beta coronavirus, is accountable for an ongoing outbreak of coronavirus disease 2019 (COVID-19) [1,2], which constitutes an enormous global burden on society. COVID-19 has resulted in over 224 million confirmed cases and over 4.68 million deaths globally. The research and development of anti-COVID-19 vaccine is currently ongoing; moreover, controlling disease transmission requires the development of effective drugs to cure it.

Idiopathic pulmonary fibrosis (IPF) is a chronic progressive disease with an irreversible advanced lung failure. IPF patients suffer from lung function decline, respiratory failure, and ultimately death [3]. The risk factors for IPF and COVID-19 are reported to be common [4]. However, the molecular mechanism underlying a crosstalk between COVID-19 and IPF was poorly defined. Identification of novel molecular targets has thus become imperative for the advancement of targeted therapy for COVID-19 with antifibrotic strategies.

The goal of the current study was to investigate the potential role of differentially expressed genes (DEGs) in the association between COVID-19 and IPF. We performed an overlap of DEGs between two the diseases on a basis of 5 datasets, followed by distinguishing the upregulated and downregulated genes. Based on a series of enrichment analysis, we interpreted the function of upregulated and downregulated DEGs. Furthermore, we carried out a protein-protein interaction (PPI) network analysis in which 22 upregulated hub genes and 11 downregulated hub genes were identified. Then, we analyzed the prominent function of 22 hub genes, and it was revealed that these hub genes upregulated in COVID-19 and IPF were involved in cytokine mediation, such as cell response to interferon. Finally, we performed a drug-target analysis and 155 candidate drugs targeting upregulated hub genes were identified. The workflow of the current study is shown in Fig 1. Herein, our findings demonstrated that hub gene and the candidate drug will be beneficial to the COVID-19 treatment. We also provide an insight that we can design and develop a candidate drug against virus variant such as Delta SARS-CoV-2, when there are common risk factors between a different disease and that caused by Delta.

The high-throughput data of SARS-CoV-2 infection was obtained from biopsy of a COVID-19 patient in GSE147507. The data of IPF was obtained from biopsy of IPF patient in five datasets, including GSE13065, GSE110147, GSE1i01286, GSE53845, and GSE24206. Venn diagram was used to reveal overlapped DEGs. The magenta circle represents DEGs in GSE147507 and yellow one represents DEGs in IPF datasets. Subsequently, common DEGs were subjected to a series of enrichment analysis and PPI network investigation. Based on an identification of highly expressed hub genes, a candidate drug was predicted to be available for a crosstalk between COVID-19 and IPF during the COVID-19 therapy.

## Materials and methods

### The collection of databases and the identification of DEGs

DEGs were obtained from six datasets in Gene Expression Omnibus (GEO, https://www.ncbi.nlm.nih.gov/geo/) database [5,6] The DEGs related to SARS-CoV-2 were obtained from GSE147507 including SARS-CoV-2 infection in lung epithelium and lung alveolar cells of humans in Apr 07, 2021 [7,8]. Five GEO datasets were collected to obtain the DEGs related to IPF, including GSE13065 with 3 IPF samples and 3 normal samples lastly updated in May 02, 2019 [9], GSE110147 with 22 IPF samples and 11 normal samples lastly updated in Aug 19, 2018 [10], GSE101286 with 12 IPF samples and 3 normal samples lastly updated in Jul 25, 2021 [11], GSE53845 with 40 IPF samples and 8 normal samples lastly updated in Jan 23, 2019 [12],

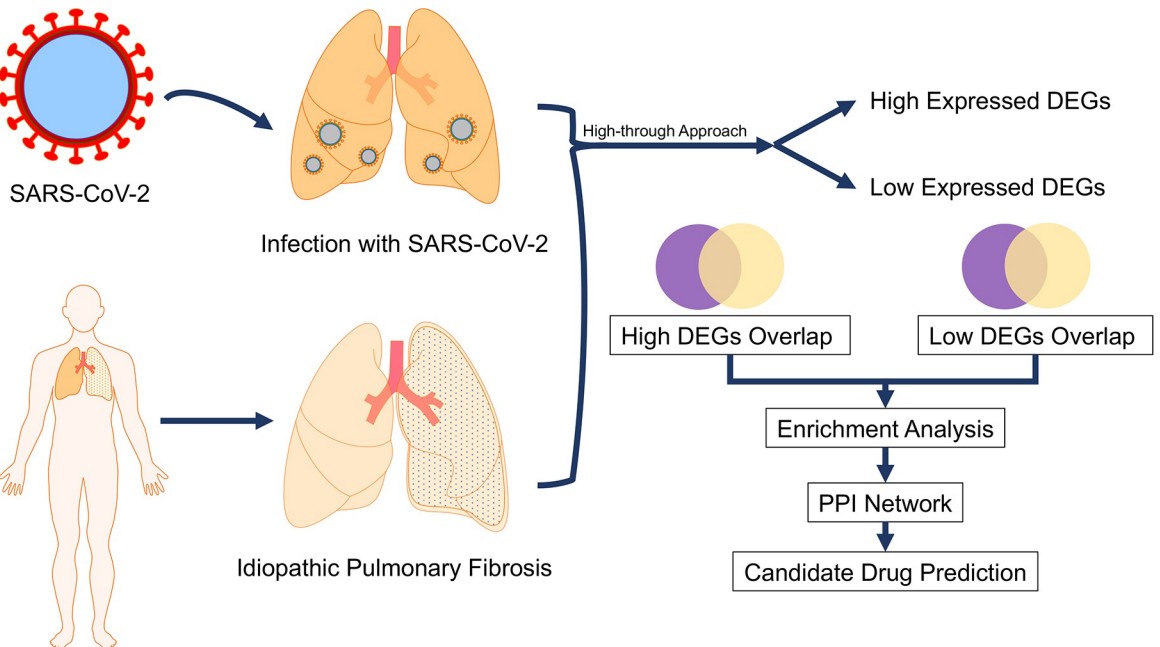

**Fig 1. The workflow of the current study.**

and GSE24206 with 17 IPF samples and 6 normal samples lastly updated in Mar 25, 2019 [13]. DEGs for the datasets were analyzed through GEO2R (https://www.ncbi.nlm.nih.gov/geo/geo2r/) web tool which uses limma package for identifying DEGs and visualized by ggplot 2 in R package. Benjamini-Hochberg method was applied to both the datasets for controlling of false discovery rate (FDR). Cut-off criteria was obtained for GSE147507 using adjusted P-value < 0.05 and log2-fold change (absolute) > 1.0. All data generated or analyzed during this study are included in this published article and its supplementary information files.

## Identification of common genes between COVID-19 and IPF

To determine detailed information of DEGs, these genes were further divided by aberrant expression level in distinct database. The adjusted P-value < 0.05 and log2-fold change > 1.0 is used as cut-off criteria for high expression DEGs, and adjusted P-value < 0.05 and log2-fold change < −1.0 for low expression DEGs in distinct dataset.

The upregulated as well as downregulated DEGs in GSE147507 were overlapped with other five datasets related to IPF.

## Enrichment analysis for common DEGs

To understand a functional characteristic of DEGs in COVID-19 and IPF, a series of enrichment analysis were adopted to gain a detailed information of biological function and pathways. Gene Ontology (GO) was performed to provide three terms, including biological process, molecular function, and cellular component [14]. Kyoto Encyclopedia of Genes and Genomes (KEGG) was used to identify metabolic pathway [15]. An online tool Enrichr (https://amp.pharm.mssm.edu/Enrichr/) was carried out to enrich the significant pathways, including WikiPathways, Reactome, and BioCarta database [16,17]. Based on the enrichment analysis, we concentrated on biological function of DEGs in both COVID-19 and IPF.

## PPI network analysis for the identification of hub genes

For assessing an association between DEGs, we established a PPI network on the Search Tool for the Retrieval of Interacting Genes (STRING) (https://string-db.org/) [18], which was utilized to predict physical and functional associations between proteins. Subsequently, we determined hub genes via an analysis of Cytohubba and MCODE on Cytoscape (3.8.2). Cytohubba (http://apps.cytoscape.org/apps/cytohubba) is a plugin of Cytoscape to explore protein associations according to topological algorithms. Top 10 hubba node was set to obtain the hub genes from DEGs. Molecular Complex Detection (MCODE) (http://apps.cytoscape.org/apps/mcode) is another plugin to provide clusters of subnetworks. The parameter of MCODE is Degree Cutoff = 2, Node score cutoff = 2, and K-score = 2.

## Prediction of candidate drugs for hub genes

The final stage of the study was designed to determine candidate drug for highly expressed hub genes. The access of the Drug Signatures database (DSigDB) is acquired through Enrichr (https://amp.pharm.mssm.edu/Enrichr/) platform, which contains the largest number of drugs/compound-related gene sets to date, were extracted and compiled from quantitative inhibition data of drugs/compounds from a variety of databases and publications [19]. Enrichr is mostly used as an enrichment analysis platform that represents numerous visualization details on collective functions for the genes that are provided as input. We predicted candidate drug targeted hub gene. The adj.P.value < 0.01 was considered statistically significant. The candidate drugs can be sorted by adj.P.value and combined score ranking.

# Results

## Identification of common DEGs between COVID-19 and IPF

From GSE147507 dataset, we identified 812 DEGs including 396 upregulated genes and 417 downregulated genes (Fig 2). Out of 5977 DEGs identified from five IPF GEO datasets, 2369 were upregulated and 3608 were downregulated. We then overlapped DEGs from one SARS-CoV-2-infected sample dataset and five IPF datasets. A total of 117 and 71 genes were identified as common upregulated (Fig 3A) and downregulated DEGs (Fig 3B), respectively. Next, we tried to identify the function of common DEGs involved during the progression of COVID-19 and IPF.

## GO and pathway identification by gene set enrichment analysis

To further understand the function and pathways of common DEGs, enrichment analysis was preformed to show that common upregulated DEGs in COVID-19 and IPF were involved in cytokine mediation, such as cell response to interferon (Fig 4, Table 1). The DEGs downregulated in both the diseases, however, were associated in the disassembly of cellular components and extracellular matrix (Fig 5, Table 2). The upregulated DEGs were mainly located in cellular component, and main molecular function of these genes was found to bind to small molecules and metabolites. Among upregulated DEGs, 25 genes were involved in cytokine-mediated signaling pathway and 11 genes were involved in cellular response to type I interferon and 11 genes in type I interferon signaling pathway in GO terms. The downregulated DEGs located at the intracellular membrane organelle and nucleus, appeared to bind with RNA and catalytic enzymes, and mediate channel activity. The GO results were consistent with those of a serial pathway analysis, including KEGG, Reactome, wikipathway, and Biocarta. For instance, Reactome and wikipathway revealed that upregulated DEGs were related to interferon signaling

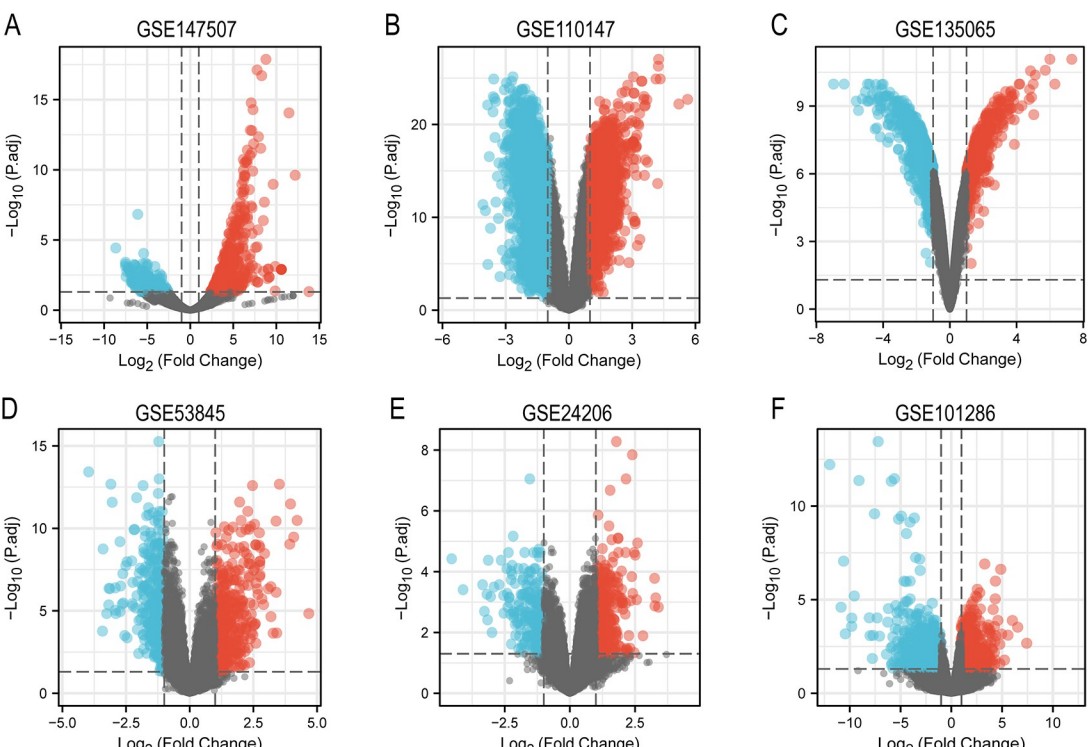

**Fig 2. Volcano plot of differentially expressed gene from COVID-19 patient samples and IPF patient samples.** A: Volcano plot of differentially expressed genes between uninfected human lung biopsies and that of deceased COVID-19 patient. B-F: Volcano plot of differentially expressed genes between lung samples from IPF patients and healthy control. The red plots with | log FC| > 1 and P value < 0.05 represent upregulated genes and blue represents downregulated genes.

and cytokine signaling pathway (S1 Fig), and downregulated DEGs were involved in MMP activation and collagen degradation (S2 Fig). Overall, these results indicated that upregulated and downregulated DEGs influenced entirely different biological functions.

## Identification of hub genes via PPI network analysis

The PPI network analysis revealed an association of DEGs between COVID-19 and IPF. Among 117 upregulated DEGs, 22 hub genes were identified on the basis of Cytohubba and MCODE analysis, including MX1, CCL2, CXCL10, TYROBP, STAT1, S100A12, IRF7, IL1B, TREM1, SPI1, UBE2L6, IFI44L, XAF1, IRF9, EPSTI1, ISG15, OASL, IFITM1, CMPK2, IFI6, OAS2, IFITM3. Among 71 downregulated DEGs, 11 hub genes were identified, including HMOX1, PPIG, MPHOSPH10, GNL2, MMP1, GADD45A, UTP6, TSR1, CCND1, PRMT1, URB1 (Fig 6).

## Enrichment analysis of hub genes

The GO enrichment analysis revealed that the 22 hub genes were upregulated in cellular response to type I interferon and type I interferon signaling pathway. The analysis also exhibited significant involvement of mitochondrial envelope and adenylyl transferase activity in the upregulated group (S3 Fig). In the downregulated group, 11 hub genes mostly enriched in nucleolus and nuclear lumen, were appeared to be evolved in RNA binding and mitotic G1 DNA damage checkpoint signaling (S4 Fig).

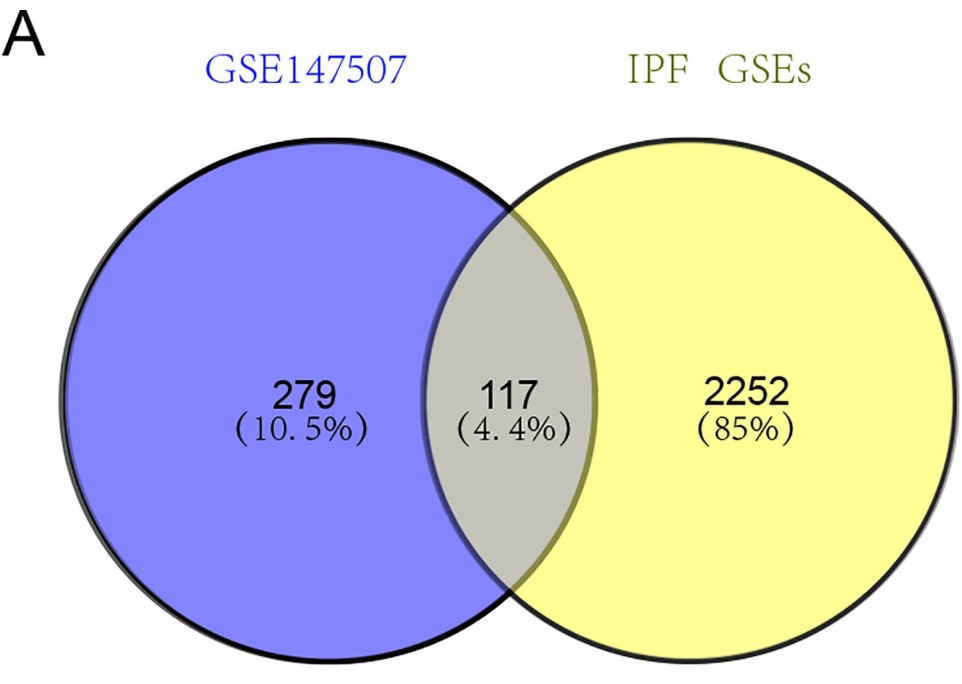

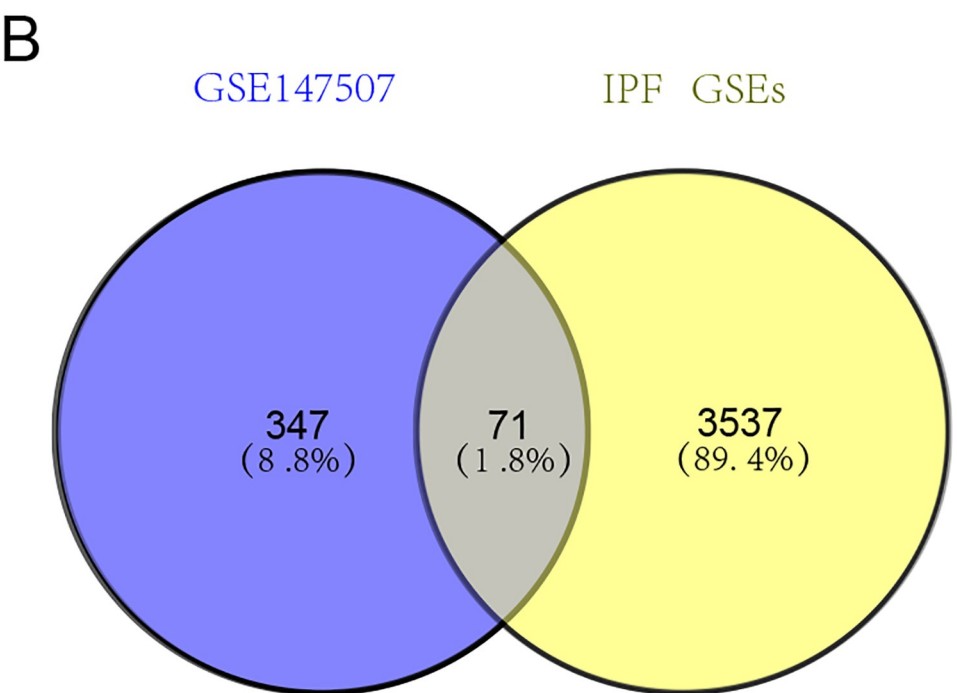

**Fig 3. Venn diagram showing differentially expressed genes between COVID-19 and IPF.** A: Upregulated DEGs between COVID-19 and IPF. B: Downregulated DEGs between COVID-19 and IPF. The blue circle in Venn diagram represents DEGs in COVID-19 dataset, and yellow circle represents five DEGs in IPF datasets.

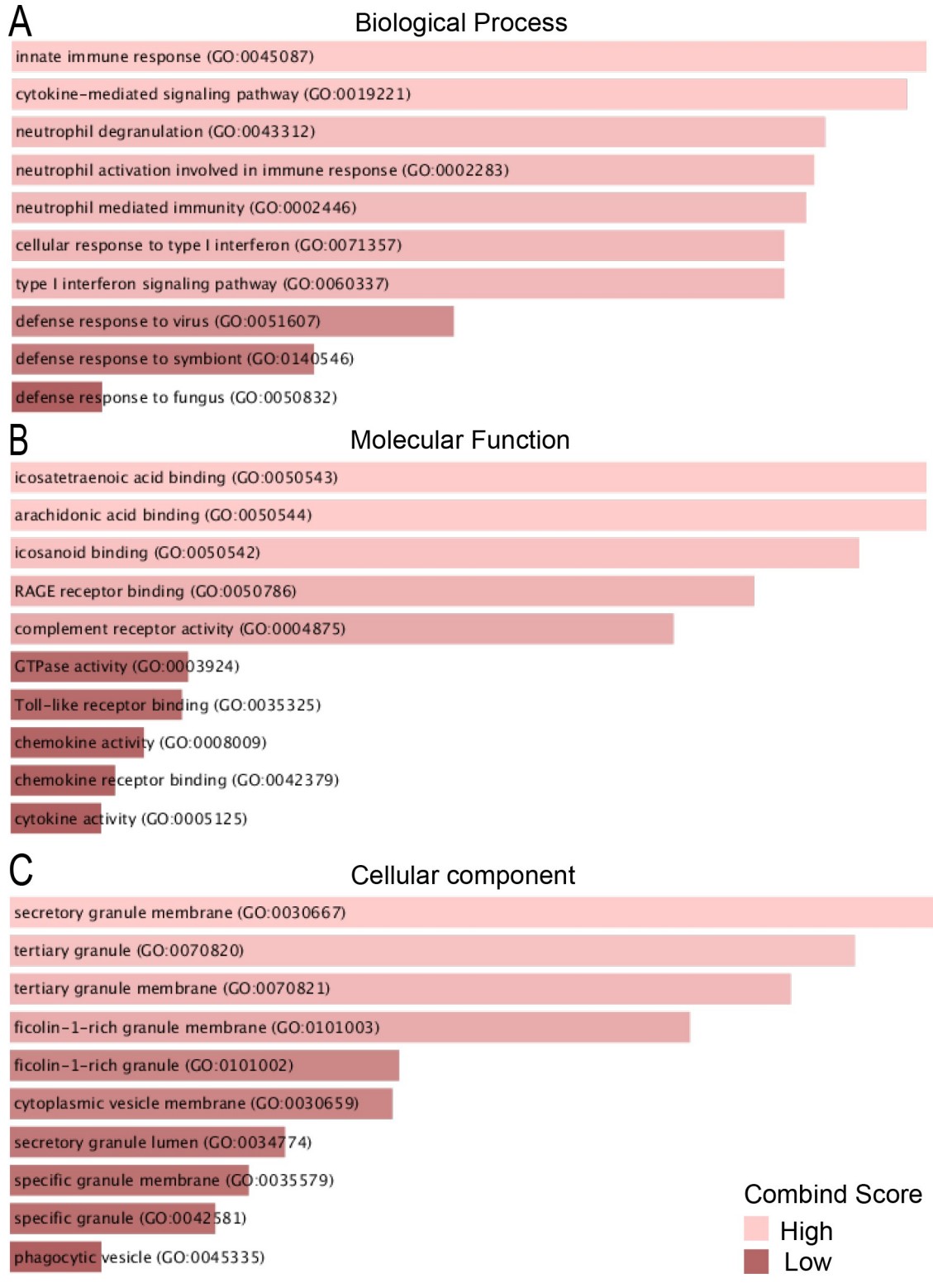

**Fig 4. GO terms of upregulated DEGs between COVID-19 and IPF.** A: GO analysis of upregulated DEGs related to biological process. B: GO analysis of upregulated DEGs related to molecular function. C: GO analysis of upregulated DEGs related to cellular component.

**Table 1. Go enrichment analysis of upregulated DEGs between COVID-19 and IPF.**

| Category | GO ID | GO Pathways | P-values | Genes |
|---|---|---|---|---|
| GO biological process | GO:0045087 | innate immune response | 9.387041929923436E-15 | IFITM3;IFITM1;CR1;FCER1G;GCH1;MX1;IFI6;ISG15;RNASE2;TREM1;SIRPB1;CXCL16;CXCL10;CLEC4D;TYROBP;CLEC7A;IRF7;S100A12;CLEC4E |
| | GO:0019221 | cytokine-mediated signaling pathway | 1.1506069546074619E-14 | IFITM3;IFITM1;CSF3R;SPI1;FPR1;IFI6;IL27;IL2RG;OASL;CA1;CCL2;HLA-DRB5;FCER1G;STAT1;IL1R2;MX1;ISG15;CXCL10;IL1A;OAS2;IL1B;IRF7;PELI1;XAF1;IRF9 |
| | GO:0043312 | neutrophil degranulation | 4.286470567853142E-14 | MGAM;CR1;FCER1G;GCA;GMFG;FPR1;GPR84;FPR2;RNASE2;MMP8;SIRPB1;PLAC8;MMP25;CLEC4D;TYROBP;SELL;S100A12;OLR1;CYSTM1;S100A9;S100A8;SIGLEC5 |
| | GO:0002283 | neutrophil activation involved in immune response | 5.071176119881218E-14 | MGAM;CR1;FCER1G;GCA;GMFG;FPR1;GPR84;FPR2;RNASE2;MMP8;SIRPB1;PLAC8;MMP25;CLEC4D;TYROBP;SELL;S100A12;OLR1;CYSTM1;S100A9;S100A8;SIGLEC5 |
| | GO:0002446 | neutrophil mediated immunity | 5.746693830188601E-14 | MGAM;CR1;FCER1G;GCA;GMFG;FPR1;GPR84;FPR2;RNASE2;MMP8;SIRPB1;PLAC8;MMP25;CLEC4D;TYROBP;SELL;S100A12;OLR1;CYSTM1;S100A9;S100A8;SIGLEC5 |
| | GO:0071357 | cellular response to type I interferon | 9.61002949577372E-14 | IFITM3;IFITM1;OAS2;STAT1;MX1;IRF7;IFI6;ISG15;XAF1;IRF9;OASL |
| | GO:0060337 | type I interferon signaling pathway | 9.61002949577372E-14 | IFITM3;IFITM1;OAS2;STAT1;MX1;IRF7;IFI6;ISG15;XAF1;IRF9;OASL |
| | GO:0051607 | defense response to virus | 1.5573135651590148E-11 | IFITM3;CXCL10;IFITM1;OAS2;STAT1;MX1;IRF7;IFI6;ISG15;RNASE2;IFI44L;OASL |
| | GO:0140546 | defense response to symbiont | 1.3693191748252762E-10 | IFITM3;IFITM1;OAS2;STAT1;MX1;IRF7;IFI6;ISG15;RNASE2;IFI44L;OASL |
| | GO:0050832 | defense response to fungus | 3.9184743402451796E-9 | CLEC4D;CLEC7A;S100A12;CLEC4E;S100A9;S100A8 |
| GO Molecular Function | GO:0030667 | secretory granule membrane | 6.097577975166167E-10 | MGAM;CR1;FCER1G;FPR1;GPR84;FPR2;SIRPB1;MMP25;CLEC4D;TYROBP;SELL;OLR1;CYSTM1;SIGLEC5 |
| | GO:0070820 | tertiary granule | 2.7444531601972747E-9 | MGAM;CLEC4D;CR1;FCER1G;FPR1;OLR1;CYSTM1;GPR84;FPR2;MMP8;SIGLEC5 |
| | GO:0070821 | tertiary granule membrane | 9.197680245668367E-9 | MGAM;FCER1G;CLEC4D;OLR1;CYSTM1;FPR2;GPR84;SIGLEC5 |
| | GO:0101003 | ficolin-1-rich granule membrane | 5.8340251207518634E-8 | MGAM;CR1;FCER1G;CLEC4D;FPR1;FPR2;SIGLEC5 |
| | GO:0101002 | ficolin-1-rich granule | 1.1269234942705415E-5 | MGAM;CLEC4D;CR1;FCER1G;GMFG;FPR1;FPR2;SIGLEC5 |
| | GO:0030659 | cytoplasmic vesicle membrane | 1.226772047106552E-5 | PSENEN;HLADRB5;TYROBP;CR1;SELL;NCF4;FPR1;IRF7;LY96;SIRPB1;SIGLEC5 |
| | GO:0034774 | secretory granule lumen | 8.887911522711858E-5 | PLAC8;SRGN;GCA;GMFG;S100A12;RNASE2;MMP8;S100A9;S100A8 |
| | GO:0035579 | specific granule membrane | 1.8064502767407338E-4 | MMP25;CLEC4D;OLR1;FPR2;GPR84 |
| | GO:0042581 | specific granule | 3.2742843178608567E-4 | MMP25;CLEC4D;OLR1;GPR84;FPR2;MMP8 |
| | GO:0045335 | phagocytic vesicle | 0.002661302293846886 | GNLY;NCF4;RAC2;CLEC4E |

*(Continued)*

**Table 1.** (Continued)

| Category | GO ID | GO Pathways | P-values | Genes |
|---|---|---|---|---|
| GO Cellular Component | GO:0030667 | secretory granule membrane | 6.097577975166167E-10 | MGAM;CR1;FCER1G;FPR1;GPR84;FPR2; SIRPB1; MMP25;CLEC4D;TYROBP;SELL;OLR1; CYSTM1; SIGLEC5 |
| | GO:0070820 | tertiary granule | 2.7444531601972747E-9 | MGAM;CLEC4D;CR1;FCER1G;FPR1;OLR1; CYSTM1;GPR84;FPR2;MMP8;SIGLEC5 |
| | GO:0070821 | tertiary granule membrane | 9.197680245668367E-9 | MGAM;FCER1G;CLEC4D;OLR1;CYSTM1; FPR2;GPR84;SIGLEC5 |
| | GO:0101003 | ficolin-1-rich granule membrane | 5.8340251207518634E-8 | MGAM;CR1;FCER1G;CLEC4D;FPR1;FPR2; SIGLEC5 |
| | GO:0101002 | ficolin-1-rich granule | 1.1269234942705415E-5 | MGAM;CLEC4D;CR1;FCER1G;GMFG;FPR1; FPR2;SIGLEC5 |
| | GO:0030659 | cytoplasmic vesicle membrane | 1.226772047106552E-5 | PSENEN;HLADRB5;TYROBP;CR1;SELL;NCF4; FPR1;IRF7;LY96;SIRPB1;SIGLEC5 |
| | GO:0034774 | secretory granule lumen | 8.887911522711858E-5 | PLAC8;SRGN;GCA;GMFG;S100A12;RNASE2; MMP8;S100A9;S100A8 |
| | GO:0035579 | specific granule membrane | 1.8064502767407338E-4 | MMP25;CLEC4D;OLR1;FPR2;GPR84 |
| | GO:0042581 | specific granule | 3.2742843178608567E-4 | MMP25;CLEC4D;OLR1;GPR84;FPR2;MMP8 |
| | GO:0045335 | phagocytic vesicle | 0.002661302293846886 | GNLY;NCF4;RAC2;CLEC4E |

## Candidate drug prediction for targeting hub genes between COVID-19 and IPF

For further investigating the significant role of common hub genes, candidate drugs targeting the 22 upregulated hub genes were predicted (Table 3). A total of 155 candidate drugs were identified with adj.P.value < 0.01 (S3 Table). These drugs were further examined to affect molecular activity of 22 hub genes and their downstream molecules, which are displayed as a list (S4 Table). Among these drugs, 11 were predicted to target more than 10 hub molecules, while 69 drugs targeted less than 3 hub molecules.

## Discussion

A strong association between COVID-19 and IPF has been previously reported [4,20,21], and IPF was reported as risk factor for COVID-19 [4]. On the contrary, anti-fibrosis therapies are available for inhibiting severe COVID-19 progression [4]. Moreover, COVID-19 has changed the approach to treat IPF patients, since SARS-CoV-2 infection is reported to impact the prognosis of IPF patients [22]. The relevance between COVID-19 and IPF is supposed to be through the association between up- and downregulated genes. One COVID-19 dataset and five IPF datasets were analyzed, the latter are designed to analyze only the lung samples. These datasets were published from 2011 to 2019, ranging from America to East Asia to ensure that our study is broadly representative. Our finding of aberrant expressed genes from 6 GEO datasets suggested that these DEGs influenced the crosstalk between COVID-19 and IPF. In addition, the present study was designed for the identification of hub genes and the prediction of their potential drug, which may enable novel molecular targets as new COVID-19 strategies with antifibrotic treatment.

Given that common DEGs can drive the development of drugs against COVID-19 and IPF, we concentrated on the DEG-related function after dividing upregulated and downregulated

**A** Biological Process

cellular component disassembly (GO:0022411)

extracellular matrix disassembly (GO:0022617)

cellular response to UV (GO:0034644)

negative regulation of transcription by RNA polymerase II (GO:0000122)

DNA replication–independent nucleosome organization (GO:0034724)

response to UV-A (GO:0070141)

polyol biosynthetic process (GO:0046173)

mitotic G1 DNA damage checkpoint signaling (GO:0031571)

embryonic digestive tract development (GO:0048566)

DNA damage response, signal transduction by p53 class mediator (GO:0030330)

**B** Molecular Function

nucleus (GO:0005634)

intracellular membrane-bounded organelle (GO:0043231)

platelet dense tubular network membrane (GO:0031095)

platelet dense tubular network (GO:0031094)

nucleolus (GO:0005730)

nuclear lumen (GO:0031981)

platelet dense granule (GO:0042827)

intracellular non-membrane-bounded organelle (GO:0043232)

small-subunit processome (GO:0032040)

sarcoplasmic reticulum (GO:0016529)

**C** Cellular component

inositol 1,4,5 trisphosphate binding (GO:0070679)

intracellular ligand–gated ion channel activity (GO:0005217)

RNA binding (GO:0003723)

calcium–release channel activity (GO:0015278)

ligand–gated calcium channel activity (GO:0099604)

TBP-class protein binding (GO:0017025)

N–methyltransferase activity (GO:0008170)

metalloendopeptidase activity (GO:0004222)

phosphatidylinositol-3,5–bisphosphate binding (GO:0080025)

general transcription initiation factor binding (GO:0140296)

Combind Score

High

Low

**Fig 5. GO terms of downregulated DEGs between COVID-19 and IPF.** A: GO analysis of downregulated DEGs according to biological process. B: GO analysis of downregulated DEGs according to molecular function. C: GO analysis of downregulated DEGs according to cellular component.

**Table 2. Go enrichment analysis of downregulated DEGs between COVID-19 and IPF.**

| Category | GO ID | GO Pathways | P-values | Genes |
|---|---|---|---|---|
| GO biological process | GO:0022411 GO:0022617 GO:0034644 | cellular component disassembly extracellular matrix disassembly cellular response to UV | 3.6908055963859072E-6 3.6908055963859072E-6 1.6226306755965077E-4 | MMP14;MMP1;SH3PXD2B;A2M; MMP10 MMP14;MMP1;SH3PXD2B;A2M; MMP10 MMP1;XPC;CRIP1;TAF1 |
| | GO:0000122 | negative regulation of transcription by RNA polymerase II | 6.80695281487198E-4 | ZNF451;CCND1;GADD45A;ATRX; CTR9; TCF21;NR1D2;TBX2;TAF1 |
| | GO:0034724 | DNA replication-independent nucleosome organization | 9.449631056544972E-4 | NASP;ATRX |
| | GO:0070141 | response to UV-A | 0.0010999291367171904 | CCND1;MMP1 |
| | GO:0046173 | polyol biosynthetic process | 0.001443814289271468 | ITPKB;ISYNA1 |
| | GO:0031571 | mitotic G1 DNA damage checkpoint signaling | 0.001599460082312502 | CCND1;PRMT1;GADD45A |
| | GO:0048566 | embryonic digestive tract development | 0.002265165629135075 | RARRES2;TCF21 |
| | GO:0030330 | DNA damage response, signal transduction by p53 class mediator | 0.0023202515386528087 | PRMT1;GADD45A;ATRX |
| GO Molecular Function | GO:0070679 | inositol 1,4,5 trisphosphate binding | 6.693872237504711E-4 | ITPR1;ITPR3 |
| | GO:0005217 | intracellular ligand-gated ion channel activity | 6.693872237504711E-4 | ITPR1;ITPR3 |
| | GO:0003723 | RNA binding | 0.0012169581083683082 | PTCD3;UTP6;PRMT1;RNMT;DDX42; CIRBP; URB1;GNL2;MFAP1;MPHOSPH10; TSR1;PPIG; SREK1 |
| | GO:0015278 | calcium-release channel activity | 0.001632573414944103 | ITPR1;ITPR3 |
| | GO:0099604 | ligand-gated calcium channel activity | 0.0020433278549486355 | ITPR1;ITPR3 |
| | GO:0017025 | TBP-class protein binding | 0.002741371475892866 | PSMC5;TAF1 |
| | GO:0008170 | N-methyltransferase activity | 0.0029955844322773523 | PRMT1;RNMT |
| | GO:0004222 | metalloendopeptidase activity | 0.003106381363248335 | MMP14;MMP1;MMP10 |
| | GO:0080025 | phosphatidylinositol-3,5-bisphosphate binding | 0.003260434667143527 | SH3PXD2B;WIPI1 |
| | GO:0140296 | general transcription initiation factor binding | 0.003821741127686192 | PSMC5;TAF1 |
| GO Cellular Component | GO:0005634 | nucleus | 0.000147678458404248 | ZNF451;ATF6B;RNMT;DDX42;TCF21; DLST; STC1;XPC;TRAK1;CCND1;HMOX1; RALGDS;EGR1;PRMT1;GADD45A; ZNF160; ATRX;CIRBP;NR1D2;GNL2;TBX2; ITPKB; GCHFR;MMP14;PSMC5;NASP;MFAP1; HBP1; PPIG;TAF1 |
| | GO:0043231 | intracellular membrane-bounded organelle | 0.000151288413440493 | ZNF451;ATF6B;RNMT;DDX42;ITPR1; TCF21; DLST;STC1;XPC;TRAK1;CCND1; PODXL; HMOX1;RALGDS;EGR1;PRMT1; GADD45A; ZNF160;ATRX;CIRBP;NR1D2;GNL2; TBX2; ITPKB;GCHFR;MMP14;PSMC5;NASP; MFAP1; RAPGEF1;HBP1;PPIG;TAF1 |
| | GO:0031095 | platelet dense tubular network membrane | 0.000440161889353639 | ITPR1;ITPR3 |
| | GO:0031094 | platelet dense tubular network | 0.000669387223750471 | ITPR1;ITPR3 |

*(Continued)*

**Table 2.** (Continued)

| Category | GO ID | GO Pathways | P-values | Genes |
|---|---|---|---|---|
| | GO:0005730 | nucleolus | 0.00110914228877878 | SELENBP1;UTP6;PODXL;MPHOSPH10;TSR1;XPC;URB1;GNL2;TAF1 |
| | GO:0031981 | nuclear lumen | 0.00124200795340088 | SELENBP1;UTP6;PODXL;MPHOSPH10;TSR1;XPC;URB1;GNL2;TAF1 |
| | GO:0042827 | platelet dense granule | 0.00249787275567805 | RARRES2;ITPR1 |
| | GO:0043232 | intracellular non-membrane-bounded organelle | 0.00752249102119212 | SELENBP1;UTP6;PODXL;SH3PXD2B;MPHOSPH10;TSR1;XPC;URB1;GNL2;TAF1 |
| | GO:0032040 | small-subunit processome | 0.0076371805235121 | UTP6;MPHOSPH10 |
| | GO:0016529 | sarcoplasmic reticulum | 0.0111473609254622 | ITPR1;ITPR3 |

genes. Except for immune response and defense response to virus, it is somewhat surprising that upregulated DEGs are enriched in inflammatory molecules, especially cytokine-related function. Type I interferon signaling pathway and cytokine-mediated signaling pathway were mainly related to upregulated DEGs. An association between type I interferon and IPF has been reported to show that type I interferon pathway may drive chronic inflammation and fibrosis [23]. Type I interferon response was amplified based on ex vivo evidence of IPF [24]. It has been reported that there were similar cytokine profiles in IPF and COVID-19 [22], which is consistent with an observation that the level of profibrotic mediators in COVID-19 patients was increased at the serum level. Our finding was an important evidence to support an antifibrotic therapy for COVID-19 patients by mediating cytokine signaling.

In case of downregulated genes between COVID-19 and IPF, the biological function was enriched in disassembly of cellular components and extracellular matrix. The pathological changes in IPF developed from an alteration of extracellular matrix, which can replace the healthy lung tissue, contributing to the deterioration of lung compliance [25]. The lung architecture is destructed due to the secretion of excessive amounts of extracellular matrix from fibroblast and myofibroblast foci [26]. Our findings were in accordance with the previous research. In our study, matrix metalloproteinases (MMPs) which accounted for disassembly of extracellular matrix were downregulated in both COVID-19 and IPF. These findings may help us to understand that absence of these genes in COVID-19 patients might induce the progression to fibrosis.

The common hub genes between COVID-19 and IPF were the most strongly associated among all DEGs. The hub genes were indeed relevant with IPF progression. For example, our study revealed that several hub genes were related to interferon signal pathway, which was demonstrated to influence IPF treatment. Besides, 19 hub genes are involved in the enrichment of chemokines. Previous research showed that chemokine CCL2 and its downstream pathways were the key to the development of IPF [27]. Our findings from PPI network analysis were consistent with our above functional enrichment, suggesting that these hub genes could be novel therapeutic targets between COVID-19 and IPF.

Considering that the hub genes played a vital role in a crosstalk between COVID-19 and IPF, we used hub genes to identify potential candidate drugs. We found several potential candidate drugs which probably contributed to the treatment of COVID-19 and IPF. Among all candidate drugs, the current study highlights the top 10 significant drugs. Among them, candidate drugs targeting exogenous invasion enabled to be an important approach along with suloctidil, which has been suggested as potential antifungal agent [28]. 3'-Azido-3'-deoxythymidine CTD 00007047 was used as an anti-viral agent and a reverse transcriptase inhibitor

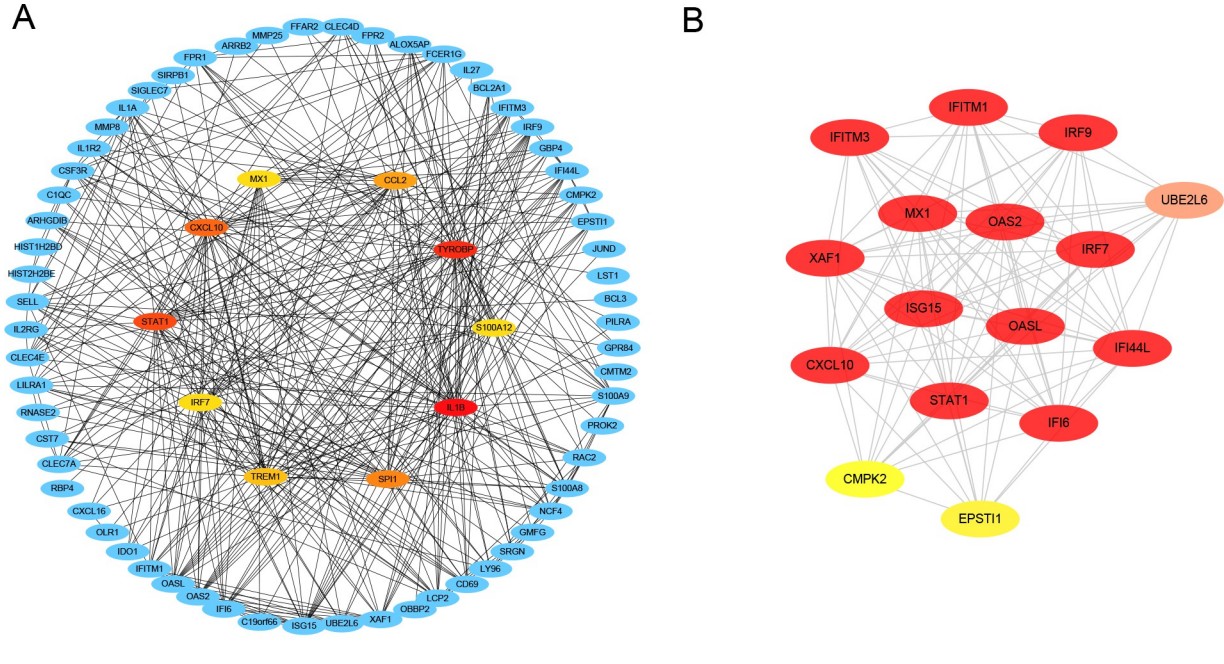

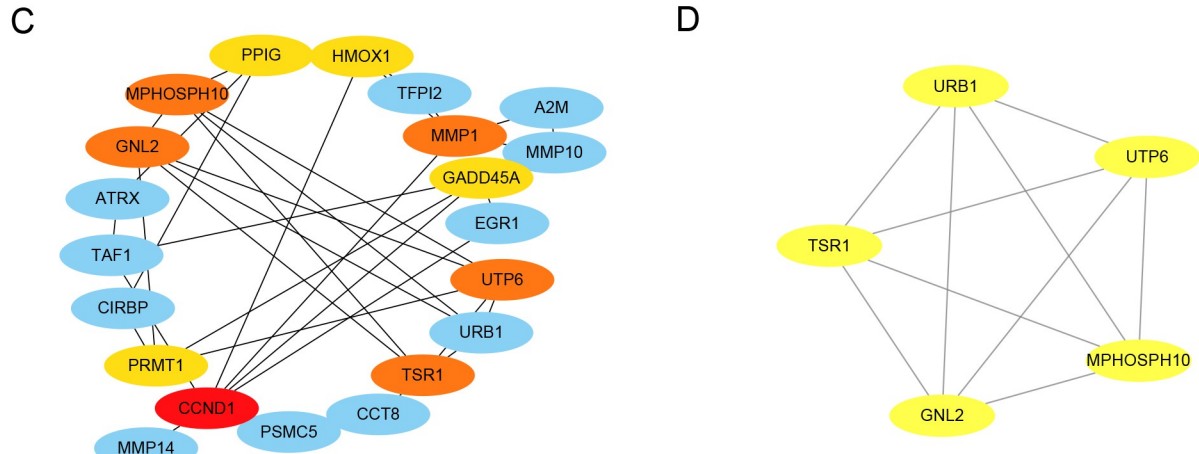

**Fig 6. Identification of hub genes from PPI network using Cytoscape plugins Cytohubba and MCODE.** A: Hub genes identified among upregulated genes using Cytohubba plugin in Cytoscape software. B: Hub genes identified among upregulated genes using MCODE plugin in Cytoscape software. C: Hub genes identified among downregulated genes using Cytohubba plugin in Cytoscape software. D: Hub genes identified among downregulated genes using MCODE plugin in Cytoscape software.

active against HIV-1, and thioridazine was proved to exhibit anti-viral activity [29]. Moreover, a previous study has revealed that myofibroblast activation and uncontrolled proliferation associated IPF with cancer [30]. Several candidate drugs exhibit anticancer activities. Chlorophyllin CTD 00000324 was determined to deactivate ERKs and inhibit breast cancer cell proliferation [31]. Prochlorperazine has been proved to exhibit anticancer activity towards different types of human cancer [32]. Terfenadine, demonstrated to be effective against PC-3 and DU-145 cells (two prostate cancer cell lines) by inducing cell apoptosis [33], and etoposide were

**Table 3. Prediction of TOP 10 candidate drugs for high expressed hub genes.**

| Name of drugs | P-value | Adjusted P-value | Genes |
|---|---|---|---|
| suloctidil HL60 UP | 1.17603903224093E-25 | 1.27835442804589E-22 | IFITM1;STAT1;MX1;IFI6;UBE2L6;ISG15;IFI44L;OASL;CXCL10;OAS2;IRF7;CCL2;XAF1;IRF9 |
| prenylamine HL60 UP | 1.06795174564932E-22 | 5.80431773760409E-20 | CXCL10;STAT1;MX1;IFI6;IRF7;ISG15;XAF1;IFI44L;IRF9;OASL |
| acetohexamide PC3 UP | 2.44772613152278E-19 | 8.86892768321756E-17 | IFITM1;STAT1;OAS2;MX1;IFI6;IFI44L;IRF9;OASL |
| chlorophyllin CTD 00000324 | 1.34723424808812E-15 | 3.66110906917947E-13 | CXCL10;IFITM1;STAT1;OAS2;MX1;IFI6;ISG15 |
| 3'-Azido-3'-deoxythymidine CTD 00007047 | 4.95534561362891E-14 | 1.07729213640292E-11 | IFITM1;STAT1;OAS2;IL1B;MX1;IFI6;IRF7;EPSTI1;CCL2;ISG15;IFI44L |
| prochlorperazine MCF7 UP | 1.95377160519666E-13 | 3.53958289141462E-11 | IFITM1;STAT1;IFI6;IRF7;ISG15;IRF9;OASL |
| terfenadine HL60 UP | 2.50627914129244E-13 | 3.89189346654984E-11 | STAT1;MX1;IFI6;IRF7;ISG15;XAF1;IRF9 |
| etoposide HL60 UP | 2.05940731534332E-12 | 2.79821968972273E-10 | STAT1;IL1B;MX1;IFI6;IRF7;CCL2;ISG15;IRF9 |
| Arsenenous acid CTD 00000922 | 8.47115116155285E-11 | 1.02312681251199E-08 | IFITM3;IFITM1;STAT1;MX1;IFI6;UBE2L6;ISG15;IFI44L;CXCL10;OAS2;IL1B;CCL2;XAF1 |
| propofol MCF7 UP | 3.01735334589948E-09 | 3.27986308699273E-07 | IFITM1;IFI6;ISG15;IRF9;OASL |

identified as anticancer drugs as they induced cancer cell apoptosis [34]. It can be assumed that candidate drugs which possess anticancer activity with the inhibition of cell proliferation and fibroblast activation might contribute to the treatment of IPF and COVID-19. In summary, the present study raised the possibility that existing drug and compounds may be available for the development of COVID-19 therapy.

Although the risk factors for IPF and COVID-19 are common, our study provides insufficient evidence to support the clinical practice of candidate drug for COVID-19 and IPF treatment. Furthermore, due to this limitation, the downstream molecules of hub genes should be determined in the future, and the role of the hub genes in crosstalk between COVID-19 and IPF should be confirmed using clinical samples and experimental models. Although the current research against COVID-19 has been conducted and data on COVID-19 in GEO are rapidly enriched, GSE147507 dataset has been verified to be reliable with solid evidence. Our conclusions were based on the responses of 5 GSEs in GEO database and so might not reflect processes via the in vivo and in vitro experiments.

## Conclusion

In summary, our results provide the common DEGs between COVID-19 and IPF, which add to the accumulating evidence that suggests a treatment for COVID-19 patients in the pulmonology ward administered antifibrotic therapy. With a series of enrichment analysis, herein, we offer new insights into the development of COVID-19 treatment on the basis of biological function. The current study unveiled a potential role of hub genes in COVID-19 and IPF, contributing to a combined COVID-19 treatment. Moreover, our findings offer some suggestions on therapeutic target identification in diseases caused by the Delta SARS-CoV-2 variant, when the common risk factor of the Delta associated with a distinct disease will be uncovered.

## Supporting information

**S1 Fig. Pathway-based enrichment analysis of upregulated DEGs between COVID-19 and IPF.** Biological entity of upregulated DEGs between COVID-19 and IPF in Wikipathways (A), KEGG (B), Reactome (C), and Biocarta (D).
(TIF)

**S2 Fig. Pathway-based enrichment analysis of downregulated DEGs between COVID-19 and IPF.** Biological entity of downregulated DEGs between COVID-19 and IPF in Wikipathways (A), KEGG (B), Reactome (C), and Biocarta (D).
(TIF)

**S3 Fig. GO and KEGG functional enrichment analysis of upregulated hub genes.** GO analysis of upregulated hub genes according to biological process (A), molecular function (B) and cellular component (C). The results of pathway terms through KEGG analysis of upregulated hub genes(D).
(TIF)

**S4 Fig. GO and KEGG functional enrichment analysis of downregulated hub genes.** GO analysis of downregulated hub genes according to biological process (A), molecular function (B) and cellular component (C). The results of pathway terms through KEGG analysis of downregulated hub genes(D).
(TIF)

**S1 Table. Pathway enrichment analysis of upregulated DEGs between COVID-19 and IPF.**
(DOCX)

**S2 Table. Pathway enrichment analysis of downregulated DEGs between COVID-19 and IPF.**
(DOCX)

**S3 Table. Prediction of candidate drugs for upregulated hub genes.**
(DOCX)

**S4 Table. The downstream molecules of 22 hub genes.**
(DOCX)

## Acknowledgments

The authors sincerely acknowledge the Helixlife (www.xiantao.love) for some bioinformatics approaches to explore the databases.

## Author Contributions

**Conceptualization:** Shilin Xia.

**Data curation:** Qianyi Chen.

**Formal analysis:** Qianyi Chen.

**Investigation:** Xueying Shi, Bingqian Huang.

**Methodology:** Qianyi Chen.

**Project administration:** Shilin Xia.

**Resources:** Xueying Shi, Bingqian Huang.

**Software:** Tingxin Wang.

**Supervision:** Hua Sui.

**Validation:** Hua Sui.

**Visualization:** Qianyi Chen.

**Writing – original draft:** Qianyi Chen.

**Writing – review & editing:** Qianyi Chen, Shilin Xia.

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
