## [Decision Letter · Decision Letter 0]

8 Nov 2021

PONE-D-21-32287Identification of hub genes associated with COVID-19 and idiopathic pulmonary fibrosis by integrated bioinformatics analysis.PLOS ONE

Dear Dr. Xia,

Thank you for submitting your manuscript to PLOS ONE. After careful consideration, we feel that it has merit but does not fully meet PLOS ONE’s publication criteria as it currently stands. Therefore, we invite you to submit a revised version of the manuscript that addresses the points raised during the review process.

We look forward to receiving your revised manuscript.

Kind regards,

Chandrabose Selvaraj, Ph.D.

Academic Editor

PLOS ONE

Journal Requirements:

Reviewers' comments:

Reviewer's Responses to Questions

5. Review Comments to the Author

Reviewer #1: Summary of the research:

The authors have aimed at identification of hub genes associated with COVID-19 and idiopathic pulmonary fibrosis (IPF) by bioinformatics analysis using publicly available databases.

Main Research Question: The authors designed their experiment to find out the hub genes upregulated in COVID-19 and IPF and subsequently predict candidate molecules against COVID-19 and IPF.

Claims: A total of 22 hub genes were reported to be upregulated in COVID-19 and IPF, for which 155 candidate molecules were predicted as potential therapeutic agents.

Conclusion of the study: The current study unveiled a potential role of hub genes

in COVID-19 and IPF, contributing to a combined treatment for COVID-19. Based on their current reports, the authors discussed on a therapeutic target identification in diseases caused by the Delta SARS-CoV-2 variant, when the common risk factor of the Delta associated with distinct disease was uncovered.

Strengths:

1. The authors have acknowledged that a strong relationship between COVID-19 and IPF has been reported in the literature and IPF was reported as risk factor for COVID-19. The antifibrosis therapies were available for inhibiting severe COVID-19 progression. The strength of the current work is in the in-depth and to a good extent exhaustive study of the upregulated hub genes in COVID-19 and IPF through database mining and computational analysis of the datasets using opensource computational tools.

2. As reported by the authors, the current study may be the first to use multiple databases to conduct study of the upregulated hub genes in COVID-19 and IPF through database mining, such as gene expression, co-expression, gene pathway enrichment analysis, etc to explore the potential molecular mechanisms of these respiratory symptoms.

Weakness: ( A few comments have been put up in the specific Major/ Minor comments for the authors along with the following points to consider. These however may not directly limit the merit of the current work. )

1. In case of all computational and public database mining and analysis the limitations of the uniformity of various datasets may pose as a limitation which have been observed here as well. Attempts by the authors to remove these are also noted.

2. The authors self-critical approach is appreciated in mentioning the perceived weakness of the study. “Although the risk factor for COVID-19 is shared with IPF, there is insufficient evidence in our analysis to support clinical practice of candidate drug for COVID-19 and IPF

treatment. Furthermore, due to the limitation, the downstream molecules of hub genes

should be determined in the future, and the role of the hub gene in crosstalk between

COVID-19 and IPF should be confirmed using clinical samples and experimental

models. Our conclusions were based on the responses of 5 GSEs in GEO database and

so might not reflect processes via the in vivo and in vitro experiment.”

Overall Recommendation: Revision recommended.

Examples and evidence:

Major issues:

1. The authors have worked in a domain of COVID-19 which still needs a lot of research and understanding. The availability of data and the fast-evolving research on the therapeutic interventions against COVID-19 makes the current work very well timed and at the same time prone to a lot of questions still unanswered. The computational analysis is always as good as the datasets. As I understand, these data on COVID-19 in GEO and other databases are rapidly enriched within very small timeframe these days. Authors comment on these lines are missing in the discussion. This may enhance the impact of the manuscript.

2. The authors self-critical approach is appreciated in mentioning the perceived weakness of the study. “Although the risk factor for COVID-19 is shared with IPF, there is insufficient evidence in our analysis to support clinical practice of candidate drug for COVID-19 and IPF

treatment.” The authors may consider, if deemed suitable, to include alternative strategies to further establish the claim for the candidate drug through support of relevant literature or relevant computational studies like DNA binding sites, binding affinity, molecular docking, MD simulations, etc.

3. The authors have also included in their discussion that “Furthermore, due to the limitation, the downstream molecules of hub genes should be determined in the future, and the role of the hub gene in crosstalk between COVID-19 and IPF should be confirmed using clinical samples and experimental models.” If the authors find it appropriate, the inclusion of the list of the downstream molecules of the 22 hub genes reported may enhance the scientific coverage of the topic aimed at in the manuscript. This will open up new avenues for researches to further enhance the knowledge gained in the current study and it will also compliment the line cited here.

4. The authors mentioned that “Our conclusions were based on the responses of 5 GSEs in GEO database and so might not reflect processes via the in vivo and in vitro experiment.” The authors may consider to add their criteria for selection/exclusion of the GSEs and establish that whether the selections were exhaustive, at least till the last date of updating of the study. If the data is exhaustive, relevant literatures in support of the claims may enhance the discussion of the manuscript.

Minor issues:

1. Page 11: The details of methods for prediction of candidate drug for hub gene is only limited to the name of the database used and the adjusted P value. The detailed criteria used for prediction, the algorithm used by the program, any adjustments applied by the authors to the default running parameters, cut-offs (apart from adjusted P value) used if any, rationale behind the choice of the specific database and its search tool, etc. may be quite useful to establish the method as reproducible as well as for clear understanding of the reader.

2. The databases, especially those on COVID-19, are rapidly getting enriched. In this context, mentioning the dates of last accession of the databases like GEO, DSigDB database, etc. This also helps the reader to understand the timeframe of the study.

3. The overall structure of the manuscript is observed to have slightly overlapping and sometimes repetitive lines or comments in methodology, results and discussion sections. The authors may consider including exclusively the points in the various sections without recurrence, for a more concise write-up.

4. Very minor but significant typographical / grammar and formatting issues were observed in the manuscript. It is assumed that a thorough proof reading by the authors during further processing will take care of these issues.

Reviewer #2: Chen et al. report an interesting analysis on common differentially expressed genes (DEGs) between individuals suffering from idiopathic pulmonary fibrosis (IPF) and COVID-19 using bioinformatics tools. In my opinion, the study is timely and deserve a publication in Plos one journal after the following comments are addressed:

1. The manuscript needs a thorough grammar check from a native English speaker as there’s random use of comma in several sentences and some sentences need to be re-framed for clarity and better understanding. For e.g., first sentence of the Abstract. There are many instances like that throughout the manuscript.

2. The last paragraph of the introduction seems redundant to me and that information is already described in the Methods section. I would suggest rewriting the last paragraph of the introduction to provide a brief overview of the study along with its findings.

3. In Fig. 1, what does magenta and yellow circles signify?

4. I think it would be better to name some of the prominent downregulated and upregulated genes in Figure 2.

4. I feel that figures 4 and 5 can be included in supplementary data.

5. Can the authors elaborate more on the role of hub genes that are common between IPF and COVID?

6. In general, the legends for the figures are less informative and should be rewritten to provide more information to the readers.

7. The details on therapeutic target identification on SARS-CoV-2 delta variant need more explanation particularly when the author highlighted that in the introduction. Could the same be applied to other VOCs?

---

## [Author Response · Author response to Decision Letter 0]

16 Dec 2021

Dear Academic Editor Chandrabose Selvaraj, 

Thank you for your letter and for the comments concerning our manuscript entitled Bioinformatic “Identification of hub genes associated with COVID-19 and idiopathic pulmonary fibrosis by integrated bioinformatics analysis”. Those comments are all valuable and very helpful for revising and improving our paper, as well as the important guiding significance to our research. We have submitted the tracked version of the manuscript, in which the revised part was colored in red in the manuscript. The authors focus on each one of comment separately and give the answers as follows:

Reviewer#1: 

Major issues:

1. The authors have worked in a domain of COVID-19 which still needs a lot of research and understanding. The availability of data and the fast-evolving research on the therapeutic interventions against COVID-19 makes the current work very well timed and at the same time prone to a lot of questions still unanswered. The computational analysis is always as good as the datasets. As I understand, these data on COVID-19 in GEO and other databases are rapidly enriched within very small timeframe these days. Authors comment on these lines are missing in the discussion. This may enhance the impact of the manuscript.

Response: Thank you for the valuable comments. It is really true as Reviewer suggested that the current research against COVID-19 is recently conducted, leading to a rapid enrichment of data on COVID-19 in GEO database. In our study, a dataset of GSE147507, published on Mar 25,2020, has been analyzed by a series of studies and has been verified to be reliable with this evidence. This dataset illustrates infections of SARS-CoV-2 in transcriptional responses and provides solid samples of human lung epithelium and lung alveolar cells not blood sample, which make this dataset more representative with a great reference value. Thank you again for this comment, and this point has been written in the paragraph six in Discussion. 

2. The authors self-critical approach is appreciated in mentioning the perceived weakness of the study. “Although the risk factor for COVID-19 is shared with IPF, there is insufficient evidence in our analysis to support clinical practice of candidate drug for COVID-19 and IPF treatment.” The authors may consider, if deemed suitable, to include alternative strategies to further establish the claim for the candidate drug through support of relevant literature or relevant computational studies like DNA binding sites, binding affinity, molecular docking, MD simulations, etc.

Response: We gratefully appreciate for your valuable suggestion, which are helpful to establish the claim for the candidate drug. In our study, the candidate drug is predicted from DSigDB database on Enrichr platform. Enrichr database is currently contains a large collection of diverse gene set libraries available for analysis and download and DSigDB gene sets were extracted and compiled from quantitative inhibition data of drugs/compounds from a variety of databases and publications, representing the direct targets of the drugs/compounds. We can provide credible candidates drug via an analysis of DSigDB as a support for clinical practice of COVID-19 and IPF.

Seven relevant literatures from Ref.28 to 34 in manuscript establish a claim for the candidate drug, providing an applicative prospect of candidate drugs in treating COVID-19 and IPF. The application of other computational studies is a great challenge for us. At present, our study is under a limitation of deeper analysis. Therefore, the online tools mentioned above provide reference for the series of prediction and offer more options for future pharmacodynamic study. Thanks again for your suggestion and this is a valuable comment for a future study to search deeper with computational methods.

3. The authors have also included in their discussion that “Furthermore, due to the limitation, the downstream molecules of hub genes should be determined in the future, and the role of the hub gene in crosstalk between COVID-19 and IPF should be confirmed using clinical samples and experimental models.” If the authors find it appropriate, the inclusion of the list of the downstream molecules of the 22 hub genes reported may enhance the scientific coverage of the topic aimed at in the manuscript. This will open up new avenues for researches to further enhance the knowledge gained in the current study and it will also compliment the line cited here.

Response: Thank you for your valuable comments. As Reviewer’s professional suggestions, we analyzed 22 hub genes based on WikiPathways platform. The downstream molecules of 22 hub genes are listed in Supplementary Table 4. 

Among 22 hub genes, the downstream molecules of three hub genes were unavailable including MX1, IFI6, and IFITM3. Hence, we illustrated their roles as follows. MX1 targeting viruses include negative-stranded RNA viruses and HBV, IFI6 negatively regulating the intrinsic apoptotic signaling pathway and TNFSF10-induced apoptosis. IFITM3 inhibits the entry of viruses to the host cell cytoplasm by preventing viral fusion with cholesterol depleted endosomes. 

We have supplied the analysis in paragraph “Candidate drug prediction for targeting hub gene between COVID-19 and IPF” in Result and the list of 22 hub gene downstream molecules are presented in Supplementary Table 4.

4.The authors mentioned that “Our conclusions were based on the responses of 5 GSEs in GEO database and so might not reflect processes via the in vivo and in vitro experiment.” The authors may consider to add their criteria for selection/exclusion of the GSEs and establish that whether the selections were exhaustive, at least till the last date of updating of the study. If the data is exhaustive, relevant literatures in support of the claims may enhance the discussion of the manuscript.

Response: We are sorry about that this part has not been fully proposed in manuscript. The Five datasets of idiopathic pulmonary fibrosis are selected to analyze the lung samples obtained from IPF patients. The datasets with blood sample, for example, are excluded because we focus mainly on lung solid tissue. And all five datasets are supported by literatures. Among them, the literature related to GSE135065 is published on Plos One, for instance.

As the reviewer concerning that if our selections are exhaustive, the five datasets are selected considering the difference of time and region. We selected GSEs published from 2011 to 2019, ranging from America to East Asia to ensure that our study is broadly representative. According to the comment, we have added some content of our criteria for selection of GSEs in the first paragraph in Discussion.

Minor issues:

1. Page 11: The details of methods for prediction of candidate drug for hub gene is only limited to the name of the database used and the adjusted P value. The detailed criteria used for prediction, the algorithm used by the program, any adjustments applied by the authors to the default running parameters, cut-offs (apart from adjusted P value) used if any, rationale behind the choice of the specific database and its search tool, etc. may be quite useful to establish the method as reproducible as well as for clear understanding of the reader.

Response: We appreciate your valuable comments. In this study, DSigDB as an online tool was used to predicted candidate drug. However, there is less parameter on this tool. For a clear understanding of the reader, detailed information of DSigDB was provided in Ref.19 in our manuscript. Readers can follow this reference to obtain the data of drugs/compound-related gene sets. DSigDB were extracted and compiled from quantitative inhibition data of drugs/compounds from a variety of databases and publications. These genes represent the direct targets of the drugs/compounds. Apart from P-value, candidate drugs are also sorted by combined score ranking and we highlighted top ten in Table 3. The combined score is performed by the Enrichr web tool, which depends on the log of the P-value and z-score. We have rewritten the paragraph “Prediction of candidate drug for hub gene” in Methods and Material. 

2. The databases, especially those on COVID-19, are rapidly getting enriched. In this context, mentioning the dates of last accession of the databases like GEO, DSigDB database, etc. This also helps the reader to understand the timeframe of the study.

Response: Thank you for pointing out this problem in manuscript. We analyzed the datasets from GEO database and predicted potential drug targets in August 2021. We have added the dates of last accession of databases in paragraph “The collection of databases and the identification of DEGs” in Methods and Material and we appreciate your valuable comments.

3. The overall structure of the manuscript is observed to have slightly overlapping and sometimes repetitive lines or comments in methodology, results and discussion sections. The authors may consider including exclusively the points in the various sections without recurrence, for a more concise write-up.

Response: We are sorry for some redundant content in manuscript. We have rewritten the parts according to Reviewer’s suggestion.

4. Very minor but significant typographical / grammar and formatting issues were observed in the manuscript. It is assumed that a thorough proof reading by the authors during further processing will take care of these issues.

Response: We are very sorry for our construction and spelling, and we have rewritten some parts of this manuscript to improve the reading fluency. The manuscript has been edited to ensure language and grammar accuracy. The Editing Certification is submitted as “Other” document.

Reviewer #2:

1. The manuscript needs a thorough grammar check from a native English speaker as there’s random use of comma in several sentences and some sentences need to be re-framed for clarity and better understanding. For e.g., first sentence of the Abstract. There are many instances like that throughout the manuscript.

Response: Thank you for this comment. The manuscript has been edited to ensure language and grammar accuracy. The Editing Certification is submitted as “Other” document.

2. The last paragraph of the introduction seems redundant to me and that information is already described in the Methods section. I would suggest rewriting the last paragraph of the introduction to provide a brief overview of the study along with its findings.

Response: Thank you for pointing out this problem in manuscript. It is true that our last paragraph of the Introduction in manuscript is overlapping with Methods. We have made correction according to the Reviewer’s comments.

3. In Fig. 1, what does magenta and yellow circles signify?

Response: We are sorry we didn’t make it clear. The magenta and yellow circles in Figure.1 are based on an analysis of Venn diagram with a data overlap between COVID-19 gene sets and five IPF gene sets. The high expression group and low expression group are depicted separately. We have rewritten the figure legend of Figure. 1 for better understanding.

4. I think it would be better to name some of the prominent downregulated and upregulated genes in Figure 2.

Response: Thank you so much for your careful check and your constructive comments. In PPI analysis, hub genes were identified from the common DEGs via two plugins including Cytohubba and MCODE. These hub genes are regarded as the prominent genes in our study, which were still identified when the volcano plots were displayed. To be frankly, we have tried several approaches to mark the differential expressed genes. At present, this study is unable to encompass this exhibition, while this is a valuable comment.

5. I feel that figures 4 and 5 can be included in supplementary data.

Response：Thank you for your valuable comments. In our manuscript, figures 4 and 5 present the result of GO functional enrichment analysis of upregulated and downregulated common DEGs between COVID-19 and IPF. It is indicated that upregulated common DEGs were mainly involved in cytokine mediation, such as cell response to interferon. In our research, we found that type I interferon pathway may drive chronic inflammation and fibrosis in IPF, and type I interferon plays an important role in lung pathology of COVID-19, which we have elaborated in our Discussion in manuscript. Hence, we presented these two figures in our manuscript for a better understanding of our results. Hopefully, our findings may help to investigate potential targets of COVID-19 and provide ideas for future research. 

6. Can the authors elaborate more on the role of hub genes that are common between IPF and COVID?

Response: Thank you for the valuable comments. We agree that the role of hub genes in our manuscript need to be elaborated. The distinct role of single hub gene is to cross talk with other genes in biological network, thus we elaborated the role of hub genes by performing functional enrichment analysis. We performed the GO/KEGG enrichment analysis of hub genes, and the results were presented as Supplementary Figure 3 and 4. We have illustrated it in the paragraph “Enrichment analysis of hub gene” in Results.

7. In general, the legends for the figures are less informative and should be rewritten to provide more information to the readers.

Response: We are sorry we didn’t make the legends for the figures detailed and we have recorrected this part in our manuscript. Thanks for your valuable comments.

8. The details on therapeutic target identification on SARS-CoV-2 delta variant need more explanation particularly when the author highlighted that in the introduction. Could the same be applied to other VOCs?

Response: Thanks for your valuable comment. As mentioned in manuscript, our study provides an insight that we can design and develop candidate drug for virus variant, such as Delta SARS-CoV-2. We aim to provide a referable approach for candidate drug prediction for other VOCs. Give that the approach was confined to sufficient data of VOCs, this research was not yet conducted. Actually, it was a valuable direction for us. Thank you again for this comment.

Once again, thank you very much for your comments and suggestions. Look forward to your reply. Please contact me if there is any question.

Your sincerely,

Shilin Xia (corresponding author) xiashilin@dmu.edu.cn

On behalf of all the authors:

Qianyi Chen, Hua Sui, Xueying Shi, Bingqian Huang, Tingxin Wang.

---

## [Decision Letter · Decision Letter 1]

5 Jan 2022

Identification of hub genes associated with COVID-19 and idiopathic pulmonary fibrosis by integrated bioinformatics analysis.

PONE-D-21-32287R1

Dear Dr. Xia,

We’re pleased to inform you that your manuscript has been judged scientifically suitable for publication and will be formally accepted for publication once it meets all outstanding technical requirements.

Kind regards,

Chandrabose Selvaraj, Ph.D.

Academic Editor

PLOS ONE

Additional Editor Comments (optional):

Reviewers' comments:

Reviewer's Responses to Questions

**Comments to the Author**

1. If the authors have adequately addressed your comments raised in a previous round of review and you feel that this manuscript is now acceptable for publication, you may indicate that here to bypass the “Comments to the Author” section, enter your conflict of interest statement in the “Confidential to Editor” section, and submit your "Accept" recommendation.

Reviewer #2: All comments have been addressed

Reviewer #3: All comments have been addressed

6. Review Comments to the Author

Reviewer #2: (No Response)

Reviewer #3: Authors have revised the whole manuscript as per reviewer comments and in present form, it can be accepted

---

## [Editor Report · Acceptance letter]

10 Jan 2022

PONE-D-21-32287R1 

Identification of hub genes associated with COVID-19 and idiopathic pulmonary fibrosis by integrated bioinformatics analysis 

Dear Dr. Xia:

I'm pleased to inform you that your manuscript has been deemed suitable for publication in PLOS ONE. Congratulations! Your manuscript is now with our production department. 

Kind regards, 

on behalf of

Dr. Chandrabose Selvaraj 

Academic Editor

PLOS ONE